# Computational Detection of Extraprostatic Extension of Prostate Cancer on Multiparametric MRI Using Deep Learning

**DOI:** 10.3390/cancers14122821

**Published:** 2022-06-07

**Authors:** Ştefania L. Moroianu, Indrani Bhattacharya, Arun Seetharaman, Wei Shao, Christian A. Kunder, Avishkar Sharma, Pejman Ghanouni, Richard E. Fan, Geoffrey A. Sonn, Mirabela Rusu

**Affiliations:** 1Department of Applied Physics, Stanford University, Stanford, CA 94305, USA; 2Department of Radiology, Stanford University School of Medicine, Stanford, CA 94305, USA; ibhatt@stanford.edu (I.B.); weishao@stanford.edu (W.S.); avsharma@stanford.edu (A.S.); ghanouni@stanford.edu (P.G.); gsonn@stanford.edu (G.A.S.); 3Department of Urology, Stanford University School of Medicine, Stanford, CA 94305, USA; refan@stanford.edu; 4Department of Electrical Engineering, Stanford University, Stanford, CA 94305, USA; arun.s@att.net; 5Department of Pathology, Stanford University School of Medicine, Stanford, CA 94305, USA; ckunder@stanford.edu

**Keywords:** extraprostatic extension, computer-aided diagnosis, deep learning

## Abstract

**Simple Summary:**

In approximately 50% of prostate cancer patients undergoing surgical treatment, cancer has extended beyond the prostate boundary (i.e., extraprostatic extension). The aim of our study was to expand artificial intelligence (AI) models that identify cancer in the prostate to also identify the cancer that spreads outside the boundary of the prostate. By combining past models with image post-processing steps and clinical decision rules, we built an autonomous approach to detect the extension of the cancer beyond the prostate boundary using prostate MRI. Our study included 123 prostate cancer patients (38 with extraprostatic extension), and our proposed method can detect cancer outside the prostate boundary in more cases than radiologists.

**Abstract:**

The localization of extraprostatic extension (EPE), i.e., local spread of prostate cancer beyond the prostate capsular boundary, is important for risk stratification and surgical planning. However, the sensitivity of EPE detection by radiologists on MRI is low (57% on average). In this paper, we propose a method for computational detection of EPE on multiparametric MRI using deep learning. Ground truth labels of cancers and EPE were obtained in 123 patients (38 with EPE) by registering pre-surgical MRI with whole-mount digital histopathology images from radical prostatectomy. Our approach has two stages. First, we trained deep learning models using the MRI as input to generate cancer probability maps both inside and outside the prostate. Second, we built an image post-processing pipeline that generates predictions for EPE location based on the cancer probability maps and clinical knowledge. We used five-fold cross-validation to train our approach using data from 74 patients and tested it using data from an independent set of 49 patients. We compared two deep learning models for cancer detection: (i) UNet and (ii) the Correlated Signature Network for Indolent and Aggressive prostate cancer detection (CorrSigNIA). The best end-to-end model for EPE detection, which we call EPENet, was based on the CorrSigNIA cancer detection model. EPENet was successful at detecting cancers with extraprostatic extension, achieving a mean area under the receiver operator characteristic curve of 0.72 at the patient-level. On the test set, EPENet had 80.0% sensitivity and 28.2% specificity at the patient-level compared to 50.0% sensitivity and 76.9% specificity for the radiologists. To account for spatial location of predictions during evaluation, we also computed results at the sextant-level, where the prostate was divided into sextants according to standard systematic 12-core biopsy procedure. At the sextant-level, EPENet achieved mean sensitivity 61.1% and mean specificity 58.3%. Our approach has the potential to provide the location of extraprostatic extension using MRI alone, thus serving as an independent diagnostic aid to radiologists and facilitating treatment planning.

## 1. Introduction

Prostate cancer is the second most frequent cancer and the fifth leading cause of cancer death among men worldwide [1]. It is estimated that in 2022 prostate cancer will account for 268,490 new cases and 34,500 deaths in the United States alone [2]. Radical prostatectomy is a common treatment for localized prostate cancer, yet 20% to 40% of patients experience biochemical recurrence after surgery [3]. These patients are much more likely to develop metastases [4]. Positive surgical margins caused by the extension of cancer beyond the prostate gland [5], i.e., extraprostatic extension (EPE), represent a major risk factor for biochemical recurrence and reduced cancer-specific survival after radical prostatectomy [6]. Cancers with extraprostatic extension are common [7,8,9], occurring in 50% of patients undergoing radical prostatectomy [10].

Detection of extraprostatic extension may improve selection of treatment (i.e., surgery or radiation) and help with surgical planning for patients who choose surgery (radical prostatectomy). If extraprostatic extension is suspected pre-operatively, surgeons may need to resect more tissue to reduce the likelihood of positive margins. Conversely, if extraprostatic extension can be confidently ruled out, surgeons will perform nerve-sparing radical prostatectomies to reduce the risk of post-operative erectile dysfunction and urinary incontinence [11]. Current guidelines from the American Urological Association recommend multiparametric MRI imaging studies for the evaluation of extraprostatic extension [12].

The ability of radiologists to detect extraprostatic extension on MRI varies greatly (sensitivity 12–83% and specificity 63–92%) [13]. One meta-analysis concluded that, with an average sensitivity of 57% [14], MRI is not sensitive enough to reliably find tumors with extraprostatic extension. One challenge is that grading systems for reporting extraprostatic extension on MRI lack standardization [15,16]. Quantitative assessment criteria such as tumor–capsule contact line length [5,12] have been shown to be independent predictors of extraprostatic extension.

Automated approaches have the potential to improve the accuracy and reliability of extraprostatic extension detection on MRI. Prior work [17,18,19,20] primarily used radiomics to predict the presence of extraprostatic extension, with one recent study employing deep learning [7] (Table 1). These prior studies have a common set of limitations: (1) They lack spatially accurate ground truth labels, since the histopathology images and the MRI are cognitively registered, a process prone to error; (2) they solve a binary classification problem i.e., predict the presence or absence of extraprostatic extension at the index lesion level, without attempting to spatially localize extraprostatic extension; (3) none of the approaches are fully automatic, as they require radiologist input usually in the form of manual lesion segmentations (see column 3 in Table 1).

In recent years, many studies have shown promising results in automatically detecting prostate cancer on MRI using convolutional neural networks [21,22,23,24,25,26]. We argue that tumor detection inside the prostate can be viewed as a sub-task for extraprostatic extension prediction, and we propose a fully automated extraprostatic extension detection framework using these existing cancer models as building blocks. The goal of our study was not to build a better deep learning model for prostate cancer, but to evaluate the utility of existing models in detecting cancer that extends beyond the capsule.

In this paper, we propose a two-step fully automated workflow for prediction and localization of extraprostatic extension on multiparametric MRI. First, we used pre-trained deep learning models to generate cancer probability maps both inside and outside the prostate gland. Second, we defined a series of post-processing steps and clinically inspired decision rules to predict the presence or absence of extraprostatic extension for each lesion.

**Table 1 cancers-14-02821-t001:** Summary of previous studies predicting presence of extraprostatic extension on multiparametric MRI. Abbreviations: EPE = extraprostatic extension; w/ = with; CNN = convolutional neural network; SVM = support vector machine; TCL = tumor–capsule contact line length (also known as capsular contact length); PI-RADS = Prostate Imaging-Reporting and Data System; ESUR = European Society of Urogenital Radiology; ADC = apparent diffusion coefficient.

First Author (Year)	Evaluation Granularity	Radiologist Input Required	Patient Number	Method	AUC
Hou (2021) [7]	Per index lesion	Tumor segmentation	849	Radiologists CNN	0.63–0.74 0.73–0.81
Cuocolo (2021) [17]	Per index lesion	Tumor segmentation	193	Radiologists Radiomics + SVM	81–83% acc 0.73–0.80 74–79% acc
Eurboonyanun (2021) [27]	Per index	Measure TCL	95	Logistic regression w/	
	lesion			absolute TCL (euclidean)	0.80
				actual TCL (curvilinear)	0.74
Losnegard (2020) [28]	Per index lesion	Tumor segmentation	228	Radiologists Radiomics + Random forest	0.75 0.74
Park (2020) [29]	Per patient	Measure TCL	301	Radiologists using MRI-based EPE grade, ESUR score, Likert scale, TCL	0.77–0.81 0.79–0.81 0.78–0.79 0.78–0.85
Xu (2020) [19]	Per lesion (all those MRI visible)	Tumor segmentation	95	Radiomics + Regression algorithm	0.87
Shiradkar (2020) [18]	Per index lesion	Tumor and periprostatic fat segmentation	45	Radiomics + SVM	0.88
Mehralivand (2019) [30]	Per index lesion	Measure TCL	553	Logistic regression w/ MRI-based EPE grade + clinical features	0.77 0.81
Ma (2019) [31]	Per index lesion	Tumor segmentation	210	Radiologists Radiomics + Regression algorithm	0.60–0.70 0.88
Stanzione (2019) [20]	Per index lesion	Tumor segmentation	39	Radiomics + Bayesian Network	0.88
Krishna (2017) [32]	Per lesion (all those MRI visible)	Tumor segmentation	149	Radiologists Logistic regression w/ PI-RADS scores, tumor size, TCL, ADC entropy	0.61–0.67, 0.61–0.72, 0.73, 0.69, 0.76

## 2. Materials and Methods

### 2.1. Dataset

#### 2.1.1. Population Characteristics

This retrospective chart review study was approved by the Institutional Review Board of Stanford University. As a chart review of previously collected data, patient consent was waived. The study focused on a cohort of 123 patients with confirmed prostate cancer who had a pre-operative MRI and underwent radical prostatectomy at our institution. Patients were randomly split into a training and validation set (*n* = 74) and a held-out testing set (*n* = 49) (Table 2). Extraprostatic extension was present in 28 of the 74 training set patients, and 10 out of the 49 testing set patients.

#### 2.1.2. Image Acquisition

Magnetic resonance images (MRI) were acquired using 3 Tesla GE scanners (GE Healthcare, Waukesha, WI, USA) with external 32-channel body array coils without endorectal coils. The imaging protocol included T2-weighted MRI (T2w), diffusion weighted imaging (DWI), derived Apparent Diffusion Coefficient (ADC) maps and dynamic contrast-enhanced imaging sequences. Axial T2w MRI (acquired using a 2D Spin Echo protocol) and ADC maps were used in this study (see characteristics in Table 3).

The excised prostates were fixed in formalin and embedded in paraffin and then serially sectioned using customized 3D-printed molds with slice orientation and thickness matching that of T2w images, followed by staining with hematoxylin and eosin. All stained slices were scanned at 20× magnification (pixel size 0.5 μm) to generate digitized whole-mount histopathology images.

#### 2.1.3. Labels

All radical prostatectomy specimens were reviewed by a genitourinary pathologist (C.A.K.) with 10 years of experience. The expert pathologist outlined cancer, including extraprostatic extension, on all whole-mount digital histopathology slices, generating pixel-level cancer and extraprostatic extension labels. By computationally registering surgical histopathology images and the corresponding MRI, we mapped the extent of extraprostatic extension and cancer onto MRI creating pixel-level labels for both classes (details in Section 2.2.1). Prostate segmentations were available on all T2w MRI and digital histopathology slices for all patients. The prostate was segmented on T2w images by expert technologists (mean experience = 9 years) and adjusted as necessary by our expert team (C.A.K, G.S.—a urologic oncologist with 13 years of experience; P.G.—a body MR imaging radiologist with 14 years of experience; M.R.—an image analytics expert with 10 years of experience working on prostate cancer).

Additionally, all MRI in the test set were reviewed retrospectively by a radiologist (P.G. or A.S.—a body MR imaging radiologist with 5 years of experience) with the knowledge that patients had undergone prostatectomy for cancer, who indicated whether extraprostatic extension was present. These labels were used to compare our model’s performance with that of radiologists in detecting the presence of extraprostatic extension on multiparametric MRI.

### 2.2. Data Pre-Processing

#### 2.2.1. Histopathology Pre-Processing

Several pre-processing steps were applied to the digitized histopathology images:**Registration:** Each digital histopathology image was aligned with its corresponding T2w MR image using the automated affine and deformable registration method RAPSODI [33]. This enabled accurate mapping of pixel-level cancer and extraprostatic extension labels from digital histopathology images onto MRI. For details on this process, refer to [26,33].**Smoothing:** Images were smoothed with a Gaussian filter with σ=0.25 mm to avoid downsampling artifacts.**Resampling:** The Gaussian smoothed images were downsampled to an X-Y size of 224×224 pixels, resulting in an in-plane pixel size of 0.29×0.29 mm^2^.**Intensity normalization:** Each RGB channel of the resulting digital histopathology images was Z-score normalized.

#### 2.2.2. MRI Pre-Processing

Several pre-processing steps were applied to the MR images, following the procedure in [25,26]:**Affine Registration:** The T2w images and ADC images were manually registered using an affine transformation driven by the prostate segmentations on both modalities.**Resampling:** The T2w images, ADC images, prostate masks and cancer labels were projected and resampled on the corresponding histopathology images, resulting in images of 224×224 pixels, with pixel size of 0.29×0.29 mm^2^.**Intensity standardization:** We followed the procedure by Nyul et al. [34]. Using the training dataset, we learned a set of intensity histogram landmarks for T2w and ADC sequences independently. Then, we transformed the image histograms to align with the learned mean histogram of each MRI sequence. The histogram average learned in the training set was also used to align the cases in the test set. This histogram alignment intensity standardization method helps ensure similar MRI intensity distribution for all patients irrespective of scanners and scanning protocols.**Intensity normalization:** Finally, Z-score normalization was applied to the prostate regions of T2w and ADC images.

### 2.3. Proposed Approach

Our approach consists of two stages (Figure 1). First, we used deep learning models pre-trained for cancer detection to generate cancer probability maps over the entire image. Second, we defined a set of post-processing steps and heuristic rules based on clinical knowledge in order to generate localized predictions for lesions with extraprostatic extension from the cancer probability maps.

#### 2.3.1. Step 1: Deep Learning Models for Cancer Detection

Many studies focused on cancer detection inside the prostate using convolutional neural networks. The goal of our study was to evaluate the utility of these existing models in detecting cancer that has spread beyond the prostate capsule. Specifically, we implemented two networks: UNet [35] and the state-of-the-art Correlated Signature Network for Indolent and Aggressive prostate cancer detection (CorrSigNIA) [26]. Both models were pre-trained for cancer detection inside the prostate gland using a five-fold cross-validation scheme. Details on the choice and training of these models can be found in Appendix A.

At inference time, given input images of T2w and ADC axial slices $I_{T2}\left(x,y\right)$, $I_{ADC}\left(x,y\right)$ (Figure 2a,b), the pre-trained model outputs a cancer probability map for the 2D slice, $p_{Ca}\left(x,y\right)$ (Figure 2d):(1)$$p_{Ca}\left(x,y\right)=f_{\theta }\left(I_{T2},I_{ADC}\right),$$
where *x* and *y* are the pixel coordinates in the left–right and posterior–anterior directions, respectively; function $f_{\theta }$ is the convolutional neural network with learned weights θ.

#### 2.3.2. Step 2: Post-Processing Pipeline

We apply several post-processing steps to the cancer probability maps output by the deep learning models in order to generate predictions for lesions with extraprostatic extension (Figure 2):**Dilated prostate mask.** The deep learning cancer predictions become less reliable the further we look outside the prostate, since other anatomical features may drive false positives. To prevent this, we applied a dilated prostate mask to the cancer probability map. Based on the diameter of the largest extraprostatic extension lesions in our cohort, we chose to dilate the original prostate mask using kernels of size 64×64 pixels (corresponding to 1.86 cm × 1.86 cm):
(2)$$M_{pr}\left(x,y\right)=dilate\left(L_{pr}\left(x,y\right)\right),$$
where $L_{pr}\left(x,y\right)$ represents the prostate segmentation mask (pixels have value 1 inside the prostate, value 0 outside) and $M_{pr}\left(x,y\right)$ is the resulting dilated prostate mask. We set all values in the probability map outside this region to zero:
(3)$$p\left(x,y\right)=p_{Ca}\left(x,y\right)*M_{pr}\left(x,y\right),$$
where $p_{Ca}\left(x,y\right)$ is the cancer probability map output by the model (Equation (Equation 1)) (Figure 2d); ∗ denotes element-wise multiplication; the resulting p(x,y) is the 2D masked cancer probability map for a given slice (Figure 2e).**Binary threshold**. All pixels in the prediction map with probability p(x,y) greater than a fixed threshold, α, were considered to be cancer, and the rest were set to zero; α is a hyperparameter:
(4)$$p_{\alpha }\left(x,y\right)=\left\{\begin{array}{cc} p(x,y) & \mathrm{if}\;p(x,y)>\alpha \\ 0 & \mathrm{otherwise}, \end{array}\right.$$
and this results in a thresholded cancer probability map $p_{\alpha }\left(x,y\right)$ for the slice. We computed $p_{\alpha }\left(x,y\right)$ for all slices in a case, resulting in a volume $P_{\alpha }\left(x,y,z\right)$ for the patient, where *z* is the slice index.**Connected components.** Next, we computed all 3D connected components in the $P_{\alpha }\left(x,y,z\right)$ volume with connectivity value 26 using the python cc3d library [36]:
(5)$$\left\{C_{\alpha }^{\left(i\right)}\left(x,y,z\right)\right\}=\mathrm{cc}3d.\mathrm{connected}\_\mathrm{components}\left(P_{\alpha }\left(x,y,z\right)\right).$$Each component $C_{\alpha }^{\left(i\right)}\left(x,y,z\right)$ is a lesion candidate (Figure 2f). Note that the connected components function returns binary mask objects, i.e., each pixel in $C_{\alpha }^{\left(i\right)}\left(x,y,z\right)$ is either 0 or 1, and all the pixels with value 1 are connected.**Logical rules:** We used logical rules to prune these components and determine the final predictions for extraprostatic extension:**Rule I:** Component must predict cancer both inside:
(6)$$C_{\alpha }^{\left(i\right)}\cap L_{pr}\neq 0,$$
and outside the prostate capsule
(7)$$C_{\alpha }^{\left(i\right)}\setminus L_{pr}\neq 0,$$
where $L_{pr}$ is the prostate binary mask, i.e., $L_{pr}\left(x,y,z\right)=1$ for pixels inside the prostate and $L_{pr}\left(x,y,z\right)=0$ for pixels outside the prostate. If a component is either fully inside ($C_{\alpha }^{\left(i\right)}\setminus L_{pr}=0$) or fully outside the prostate ($C_{\alpha }^{\left(i\right)}\cap L_{pr}=0$), it is not a viable candidate for extraprostatic extension (Figure 2g, and brown components were discarded because they are fully outside the prostate and the green component was accepted since it crosses the prostate border).**Rule II:** For each viable lesion candidate, compute tumor–capsule contact line length (TCL) and compare with threshold $th_{TCL}$. The overlap between a candidate $C_{\alpha }^{\left(i\right)}$ and the prostate boundary $L_{pr}$ defines a curvilinear segment (shown in pink in  Figure 2g), and $l_{TCL}^{\left(i\right)}$ is the length of this segment.
(8)$$l_{TCL}^{\left(i\right)}=\mathrm{tumor}\_\mathrm{capsule}\_\mathrm{contact}\_\mathrm{line}\_\mathrm{length}\left(C_{\alpha }^{\left(i\right)},L_{pr}\right).$$Lesion candidates with $l_{TCL}^{\left(i\right)}<th_{TCL}$ are discarded. Candidates with $l_{TCL}^{\left(i\right)}\geq th_{TCL}$ constitute our final predictions for cancer lesions with extraprostatic extension. Each final candidate $C_{\alpha }^{\left(i\right)}$ is a binary mask; multiplying it element-wise with the probability map $P_{\alpha }$ gives the probability map for cancer with extraprostatic extension (Figure 2h).
(9)$$q^{\left(i\right)}\left(x,y,z\right)=C_{\alpha }^{\left(i\right)}\left(x,y,z\right)*P_{\alpha }\left(x,y,z\right).$$We denote the final extraprostatic extension probability map for the entire case volume Q(x,y,z).

Algorithm 1 presents all these steps in pseudo-code.

We refer to our proposed end-to-end model that includes pre-trained CorrSigNIA + post-processing steps + decision rules as EPENet, and we refer to the baseline end-to-end model pre-trained UNet + post-processing steps + decision rules as UNet_EPE. Both models take as input the T2w and ADC images and output the extraprostatic extension probability map volume Q(x,y,z) for the patient.
**Algorithm 1** Steps for predicting lesions with extraprostatic extension**Require:**$I_{T2},I_{ADC},L_{pr}$**Require:** user defined parameters $\alpha ,\;TCL_{th}$**Ensure:** load pre-trained cancer model $f_{\theta }$   $p_{Ca}\leftarrow f_{\theta }\left(I_{T2},I_{ADC}\right)$              ▹ deep learning model inference, Equation (Equation 1)  $M_{pr}\leftarrow$ dilate$\left(L_{pr}\right)$                                  ▹ Equation (Equation 2)  $p\leftarrow p_{Ca}*M_{pr}$                         ▹ apply dilated prostate mask, Equation (Equation 3)  $p_{\alpha }\leftarrow$ binary threshold (p; α)                            ▹ Equation (Equation 4)  $C_{\alpha }\leftarrow$ connected components $\left(p_{\alpha }\right)$                         ▹ Equation (Equation 5)  $n\leftarrow len(C_{\alpha })$   q← empty list                ▹ where final EPE lesion predictions will be appended  **for**
*i* in 0,1,...n−1**do**                       ▹ iterate over all candidates $C_{\alpha }^{\left(i\right)}$    **if** $C_{\alpha }^{\left(i\right)}\cap L_{pr}\neq 0$ and $C_{\alpha }^{\left(i\right)}\setminus L_{pr}\neq 0$ **then**               ▹ Equations (Equation 6) and (Equation 7)        $l_{TCL}^{\left(i\right)}\leftarrow$ compute tumor–capsule contact line length$\left(C_{\alpha }^{\left(i\right)},L_{pr}\right)$         ▹ Equation (Equation 8)        **if** $l_{TCL}^{\left(i\right)}>TCL_{th}$ **then**           $q\leftarrow \mathrm{append}(C_{\alpha }^{\left(i\right)}*p_{\alpha })$         ▹ component $C_{\alpha }^{\left(i\right)}$ satisfied all criteria, Equation (Equation 9)                                     ▹ append to list of final EPE predictions        **end if**    **else** continue    **end if****end for**Q← aggregate all final predictions for the case, *q*

### 2.4. Evaluation

#### 2.4.1. Patient-Level Evaluation

If there are extraprostatic extension labels on any slice in a case, the patient is labeled as ground truth EPE positive; otherwise, the patient is ground truth EPE negative. If our approach predicts extraprostatic extension on any of the slices (Q≠0), then the patient is predicted EPE positive. If Q(x,y,z)=0 on all slices, the patient is predicted EPE negative. The standard definitions for true positive, false positive, true negative and false negative predictions apply.

#### 2.4.2. Sextant-Level Evaluation

Following standard clinical practice for systematic 12-core biopsies [37,38], we computationally divide the prostate volume into six regions, i.e., sextants, corresponding to left and right side and the apex, midgland and base regions. For details on this, refer to Seetharaman et al. [25]. For each sextant, we assign ground truth labels as follows: if the sextant contains any extraprostatic extension labels, then it is ground truth EPE positive; otherwise, it is ground truth EPE negative. Consequently, ground truth EPE negative patients will have six ground truth negative sextants, and ground truth EPE positive patients will have one or more ground truth positive sextants.

If our approach predicts extraprostatic extension anywhere within a sextant (Q≠0), then that sextant is predicted EPE positive. If Q(x,y,z)=0 throughout a sextant, that sextant is predicted EPE negative. The standard definitions are used to count true positive, false positive, true negative and false negative predictions.

#### 2.4.3. ROC Analysis

To generate receiver operating characteristic (ROC) curves, we varied the binary threshold parameter α uniformly between [0,1] in steps of 0.05. For each value of α we performed the post-processing steps in Algorithm 1 and the evaluation steps at the patient and sextant levels, recording true positive and false positive rates. We then generated two ROC curves for each model: a patient-level ROC and a sextant-level ROC, each with its corresponding area under the curve (AUC) score.

### 2.5. Experimental Design

#### Hyper-Parameter Optimization

The method described in Algorithm 1 has two user-specified parameters: the binary threshold parameter α, and the tumor-capsule contact line length threshold $TCL_{th}$.

Parameter α specifies the threshold applied on the cancer probability map to create a binary mask (Equation (Equation 4)), from which connected components are computed and lesion candidates determined. By varying α we generate receiver operating characteristic curves (Section 2.4.3).

The tumor-capsule contact line length threshold $TCL_{th}$ is the length of the contact line between a lesion and the prostate boundary, above which the algorithm predicts extraprostatic extension for the lesion (Equation (Equation 8)). There is no consensus in the literature on the best tumor-capsule contact line length value to use in clinical nomograms, with proposed values ranging from 10 mm to 25 mm [39]. To find the optimal value for the $TCL_{th}$ parameter, we ran a grid search on the training data, in which we evaluated a range of thresholds and selected the one with the best performance. Grid search is a tuning technique that attempts to compute the optimum values of hyperparameters. It is an exhaustive search that is performed on a specific parameter.

For the proposed EPENet and the baseline UNet_EPE, we generated extraprostatic extension predictions at different values for $TCL_{th}\in${2.5 mm, 5.0 mm, 7.5 mm, 10.0 mm, 12.5 mm, 15.0 mm, 17.5 mm, 20.0 mm}. We performed five-fold cross-validation with respect to the training dataset of the deep learning models. We constructed the receiver operating characteristic curves and computed AUC at per-patient and per-sextant level, as described in Section 2.4. Mean AUC was computed over the validation sets of the five cross-validation folds, both at sextant-level and patient-level. Figure 3 shows the results of the grid search over parameter $TCL_{th}$ for EPENet and UNet_EPE.

Based on the graphs in Figure 3, we chose $TCL_{th}=10$ mm, since it provided the best balance between good performance at sextant-level and patient-level. All experiments in the following sections were conducted with the tumor-capsule contact line length threshold set to $TCL_{th}=10$ mm.

## 3. Results

### 3.1. Qualitative Results

We used 3DSlicer [40] to visualize EPENet predictions overlaid on top of MR images. EPENet successfully detected extraprostatic extension (representative case shown in Figure 4). Note that EPENet final outputs indicate only lesions with a predicted extraprostatic extension component. Slices on which the model predicts cancer that is fully contained within the prostate capsule will not show any EPENet final predictions. For selective identification of cancer inside the prostate gland, the CorrSigNIA model should be applied directly. EPENet is designed as a tool to specifically highlight areas at risk for extraprostatic extension and should be used accordingly.

Figure 4 shows predictions computed with tumor–capsule contact line length threshold $TCL_{th}=10$ mm and two different binary thresholds α=0.1 and α=0.3. For both values of α, EPENet generated true positive predictions at the patient-level. However, varying α changes the model performance at the sextant-level (which takes into account the spatial location of predictions). Column 1 of Figure 4 schematically illustrates the division of prostate into sextants corresponding to the apex, mid, base, right- and left-hand side regions of the gland. This case has two ground truth EPE positive sextants (spanned by the extraprostatic extension on the left side of the patient) and four ground truth EPE negative sextants. The EPENet model with α=0.3 was 100% sensitive and 100% specific, since the final prediction was confined to the two ground-truth positive sextants (apex and mid, patient left-hand side). The EPENet model with α=0.1 was 100% sensitive but only 50% specific, because the EPE prediction extended into the apex and mid sextants on the patient right-hand side, thus accounting for two false positive sextant predictions. To ensure spatially accurate predictions, it is important to select an appropriate operating point where the model balances sensitivity and specificity at the sextant-level. We address this in the next subsection.

Figure A1 in Appendix B shows qualitative results for additional cases in the test set.

### 3.2. Quantitative Results

Table 4 shows sensitivity and specificity results for EPENet operating at various binary thresholds α. We observed that for α<0.30 specificity at the sextant level dropped below 50% in the cross-validation set. Conversely, for α>0.35 sensitivity at the sextant level dropped below 50% in the cross-validation set. We recommend an optimal operating point with 0.30≤α≤0.35 to achieve the best balance between sensitivity and specificity at patient-level and sextant-level. Within this range, we selected α=0.30 to maximize sensitivity.

Figure 5 displays the receiver operating characteristic curves (ROC) for EPENet and UNet_EPE in the cross-validation folds and test set. Over the five cross-validation folds, at the patient level, EPENet had mean AUC=0.72±0.16, while UNet_EPE performed close to chance with mean AUC=0.53±0.07. At the sextant-level, EPENet (mean AUC=0.64±0.16) also outperformed the baseline UNet_EPE (mean AUC=0.50±0.06).

On the test set, the radiologists assessment of EPE was 50.0% sensitive and 76.9% specific. By comparison, EPENet was 80.0% sensitive and 28.2% specific in the test set. While our proposed method had more false positives (and hence lower specificity), it also detected extraprostatic extension missed by the radiologists. The test set included 10 patients with extraprostatic extension: Radiologists correctly identified five cases, while EPENet correctly identified eight cases. The two cases missed by EPENet had focal extraprostatic extensions associated with smaller cancer lesions inside the prostate (index lesion volumes of 640 mm^3^ and 470 mm^3^ compared to test cohort median 1100 mm^3^—see Table 2). The radiologists also missed one of these extraprostatic extensions. Out of the 10 patients with extraprostatic extension in the test set, 5 presented focal extraprostatic extension and 5 showed multifocal extraprostatic extension. EPENet correctly predicted all multifocal cases, whereas the radiologists only correctly predicted two of the multifocal cases.

## 4. Discussion

We developed and validated a fully automated pipeline for the prediction and localization of extraprostatic extension in patients with prostate cancer. We used pre-trained deep learning models to generate cancer probability maps and applied post-processing steps and clinical decision rules to arrive at final predictions for lesions with extraprostatic extension. The end-to-end EPENet model operates without any radiologist input and only requires T2w and ADC images to make a final prediction. We recommend 0.30≤α≤0.35 as the optimal operating range for this model and tumor-capsule contact line length threshold $TCL_{th}=10\;\mathrm{mm}$. For $TCL_{th}=10\;\mathrm{mm}$ and α=0.30, EPENet achieved sensitivity 80.0% and specificity 28.2% at the patient-level, compared to the radiologists 50.0% sensitivity and 76.9% specificity on test set data. When accounting for the spatial location of predictions, at the sextant-level, EPENet model had a balanced 61.1% sensitivity and 58.3% specificity. High sensitivity shows that our method can reliably detect the presence of extraprostatic extension. As such, EPENet can be useful in highlighting regions suspicious for extraprostatic extension that might otherwise be difficult to detect by the human eye. Given the low specificity at the patient-level, the intended use for EPENet would be as a detection aid tool for radiologists rather than a standalone diagnosis software. Future work will assess performance of radiologists with and without our EPENet system and investigate the best strategies to integrate the proposed application to guide radiologists during prostate MRI interpretation in clinical practice.

We note that local artifacts or other anatomical features outside the prostate capsule that appear dark on MRI can mislead deep learning models into predicting extraprostatic extension (see examples in Figure A1 rows 2 and 3). None of the prior works on EPE detection were confronted with this challenge since they used radiologist segmented lesions as the starting point for extraprostatic extension prediction. This diminishes the risk of false positives, but increases the risk of false negatives, since any lesion missed by the radiologist will be, by default, also missed by the EPE model.

Previous studies showed that the radiologists AUC for predicting extraprostatic extension ranged from 0.60 to 0.75. This is similar to our radiologists performance (AUC = 0.63). Park et al. [29] showed that by using additional metrics such as tumor–capsule contact line length or the EPE grading system proposed by Mehralivand et al. [30], radiologists performance improved, with AUC ranging from 0.77 to 0.85. Radiomics methods reported AUCs between 0.73 and 0.88, while the convolutional neural network PAGNet [7] had an AUC in the range 0.73 to 0.81. Since all the aforementioned methods are trained using radiologist cancer labels as inputs together with MR images, we cannot directly compare the reported AUC performance with that of EPENet (0.72±0.16 at patient-level). Both the radiomics methods and PAGNet required the radiologists to manually segment the index lesions on all slices, making them time-consuming methods. Furthermore, they all solve a binary classification problem at the patient-level rather than a spatial detection problem and will omit any lesion missed by the radiologists. While EPENet generates a localized prediction for extraprostatic extension, it does not predict the size of the EPE lesion. Future research will focus on distinguishing between predictions for macroscopic EPE and microscopic EPE.

The automatic detection of extraprostatic extension is a challenging task for two key reasons. First, extraprostatic extension regions are small (see Table 2, cancer lesion volumes range between 0.7–3.4 mL, while median extraprostatic extension volume is <0.01 mL) and it can even be debated whether MRI might not even capture these small regions (sometimes only representing a handful of pixels). Second, MRI intensities vary greatly across patients, making it difficult to learn features that capture very small cancers, especially since both the prostate boundary and cancer are hypointense on T2w images. Appendix C provides further details and causes for the large number of false positives.

Our study is the first to explore deep learning-based strategies that simultaneously detect tumors and assess the presence of extraprostatic extension. EPENet had an AUC of 0.64 ± 0.16 at the sextant-level (0.66 in the test set) and 0.72 ± 0.16 at the patient level (0.54 in the test set). The main strength of our method is that it operates without any human input. This has two advantages. First, EPENet brings added value by being able to highlight regions suspicious for extraprostatic extension that might otherwise be missed by the radiologists. Second, it eliminates the need for expensive and time-consuming manual annotations. Furthermore, the framework presented in this paper is fully independent of the underlying deep learning model used and, thus, can be applied out-of-the-box to any convolutional neural network trained for prostate cancer detection. As we develop better deep learning models, or ensemble models, the proposed framework can be applied to further improve extraprostatic extension detection, at minimal additional computational cost. An optimized system for EPE detection could have high impact in treatment planning, since it would enable the surgeon to consider nerve-sparing radical prostatectomy, which has been shown to significantly reduce the risk of post-operative erectile dysfunction and urinary incontinence [11,41].

Our study had two noteworthy limitations. First, it included a relatively small dataset from a single institution, with all MRIs acquired on scanners from a single manufacturer. This limits the generalization power of our model. There are also limitations in our dataset labels. Extraprostatic extension labels were defined by mapping from histology. Regions with missing histology (such as the extreme apex and extreme base of the prostate) may have missing EPE labels. We also disregarded seminal vesicle invasion. Second, we had several extraprostatic extension false positives driven by anatomical features outside the prostate that appear naturally dark on MRI. Future work will focus on increasing the cohort size and validate our framework on data from external institutions, as well as improved strategies for reducing false positives.

## 5. Conclusions

We introduced an approach to detect cancer lesions with extraprostatic extension on prostate multiparametric MRI. We used existing deep learning architectures and defined a framework to identify which candidate lesions might have extraprostatic extension. An optimized approach for EPE detection can serve to assist radiologists during their interpretation of MRI, providing an independent assessment of the presence of extraprostatic extension. Our approach may facilitate the planning of treatments, e.g., image-guided focal therapy or surgical removal of the prostate.

## Figures and Tables

**Figure 1 cancers-14-02821-f001:**
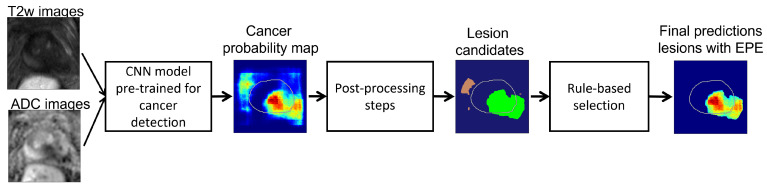
Flowchart describing our method for generating spatial predictions of cancer lesions with extraprostatic extension. A convolutional neural network takes in T2w and ADC images and outputs a probability map for cancer. We apply post-processing steps of masking, thresholding and connected components to the probability map to obtain a set of lesion candidates. We check each lesion candidate against a set of heuristic rules and determine which are the lesions with suspected extraprostatic extension.

**Figure 2 cancers-14-02821-f002:**
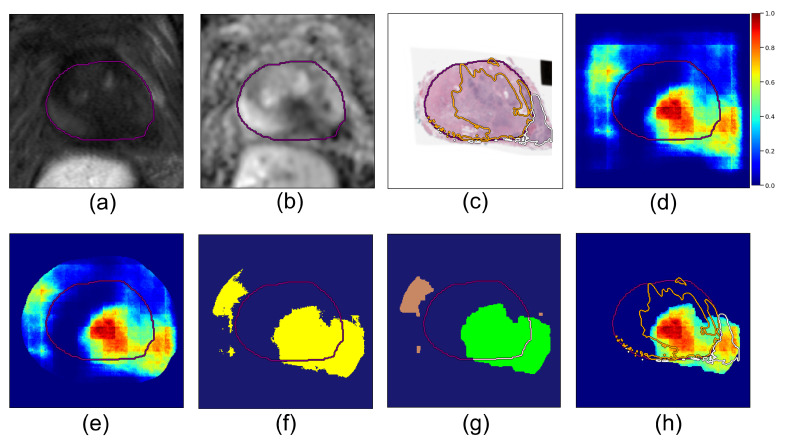
**Row 1:** Registered MRI and histopathology slices, along with output from cancer detection deep learning model in a patient with bulky extraprostatic extension. (**a**) T2w image, (**b**) ADC image, (**c**) Histopathology image overlaid with ground truth cancer labels and prostate segmentation mask. Purple contour shows prostate gland segmentation, orange contour is cancer inside the prostate, white contour is extraprostatic extension (EPE). (**d**) Cancer probability map output by the pre-trained deep learning cancer detection model. **Row 2:** Post-processing steps to generate final predictions for cancer with extraprostatic extension. (**e**) Cancer probability map after applying dilated prostate mask. (**f**) Candidate lesions (shown in yellow) were obtained by applying binary threshold and detecting connected components. (**g**) Candidate lesions were pruned based on their location relative to the prostate; components fully inside or fully outside the prostate were rejected (brown), those crossing the border were accepted (green). For the green lesion candidate, the tumor-capsule contact line (TCL) is displayed in pink. (**h**) Final prediction map for lesion with EPE.

**Figure 3 cancers-14-02821-f003:**
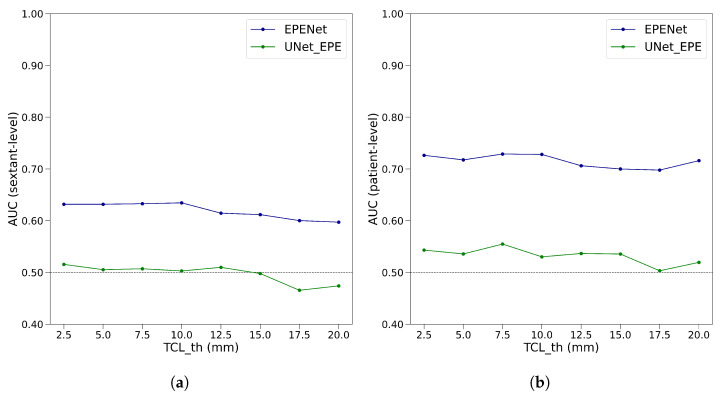
Grid search over the tumor-capsule contact line length threshold parameter ($TCL_{th}$) for EPENet and UNet_EPE. The two panels correspond to the evaluation metric used (displayed on the *y* axis): sextant-level AUC in panel (**a**) and patient-level AUC in panel (**b**). The *x* axis shows the values of $TCL_{th}$ in millimeters. Results are averages computed over the validation sets of the five cross-validation data folds.

**Figure 4 cancers-14-02821-f004:**
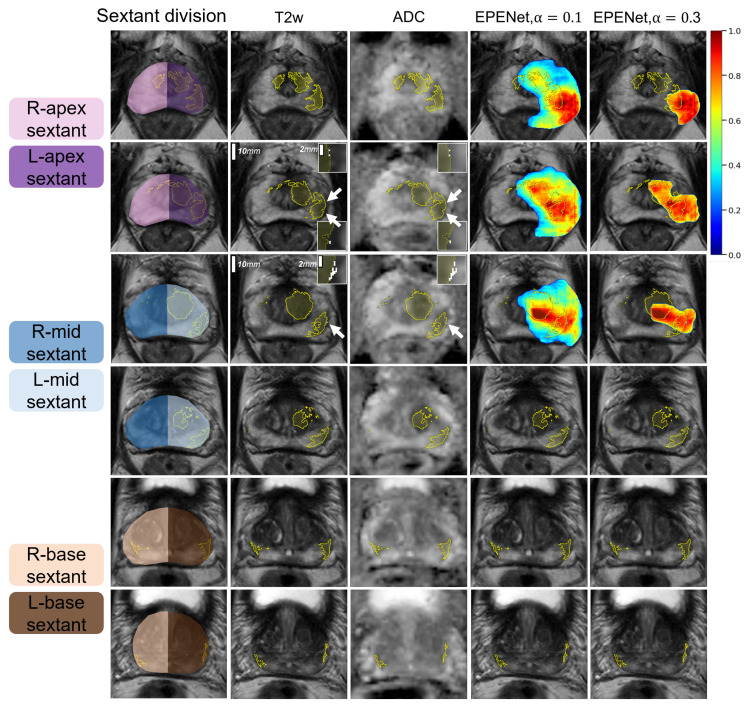
EPENet predictions for cancers with extraprostatic extension in one example case from the test set, from apex to base. EPENet does not show any predictions for cancer lesions fully contained within the prostate capsule. Column (1) Schematic illustration of the prostate division into sextant regions. Column (2) Input T2w images. Column (3) Input ADC images; yellow labels outline cancer within the prostate, white labels outline extraprostatic extension (EPE); white arrows point to the small EPE regions; the panels in the corners show zoomed-in 5 mm × 5mm areas containing EPE. This patient shows cancer on all slices, but EPE is present only in rows 2 and 3. Column (4) and (5) show EPENet predictions for different values of the α parameter. Predictions are probability maps displayed in a cold to hot color scheme (dark blue–0, dark red–1).

**Figure 5 cancers-14-02821-f005:**
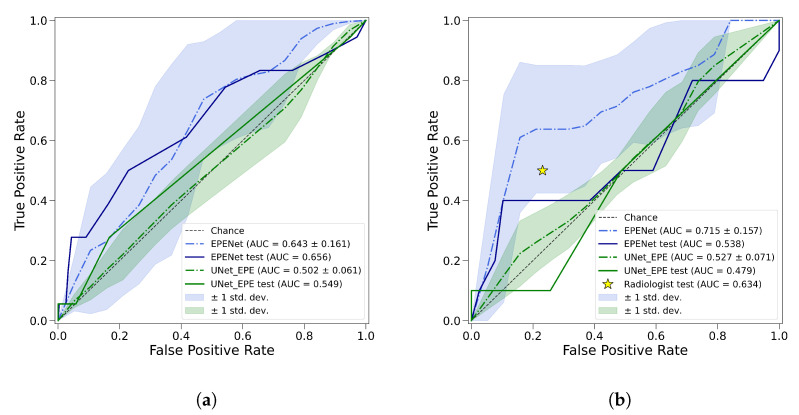
Receiver operating characteristic curves based on performance metrics at the sextant-level (**a**) and patient-level (**b**). Dashed lines are the mean performance over the five cross-validation folds and the shaded regions represent one standard deviation around the average. Solid lines are the ROC results in the test set. EPENet model is shown in blue, UNet_EPE model is green. In panel (**b**), the star marker represents radiologists performance on the test set.

**Table 2 cancers-14-02821-t002:** Details of the train and test cohorts. Lesion volumes are presented as median (IQR), where IQR = interquartile range. Lesion volumes were computed based on the histopathology labels, assuming continuity between slices and using the MRI slice thickness.

Cohort	Train	Test
Patient number	74	49
Lesion count	90	58
Indolent	9	10
Aggressive	81	48
EPE (pathologically proven)	29	10
Lesion volume (mm^3^)	1541.6 (714.7, 3418.6)	1099.1 (743.2, 2544.7)
EPE volume (where applicable)	8.6 (3.6, 44.6)	10.6 (5.6, 36.3)

**Table 3 cancers-14-02821-t003:** Radical prostatectomy cohort data characteristics. Abbreviations: T2w—T2-weighted MRI; ADC—Apparent Diffusion Coefficient map. [a,b] indicates range between *a* and *b*.

Number of Patients	123
T2w	
Repetition time (TR, range) (s)	[3.9,6.3]
Echo time (TE, range) (ms)	[122,130]
Pixel size (range) (mm)	[0.27,0.94]
Distance between slices (mm)	[3,5.2]
Matrix size	[256,512]
Number of slices	[20,44]
ADC	
b-values (s/mm^2^)	0,50,800,1000,1200
Pixel size (range) (mm)	[0.78,1.50]
Distance between slices (mm)	[3,4.5]
Matrix size	[50,256]
Number of slices	[15,40]

**Table 4 cancers-14-02821-t004:** Sensitivity and specificity results in the cross-validation and test sets for the EPENet model operating at various binary thresholds α. We recommend 0.30≤α≤0.35 as the optimal operating range for this model. Results within the recommended operating range are shown in bold.

Mode	Threshold α	Cohort	Sensitivity	Specificity	Sensitivity	Specificity
			(Patient)	(Patient)	(Sextant)	(Sextant)
			%	%	%	%
EPENet	0.10	cross-val	100.0±0.0	1.7±3.3	96.5±7.1	17.9±8.3
		test	90.0	0.0	88.9	13.0
EPENet	0.15	cross-val	100.0±0.0	11.1±6.7	95.3±9.4	29.2±9.5
		test	90.0	0.0	83.3	23.9
EPENet	0.20	cross-val	100.0±0.0	14.3±9.6	88.8±13.0	36.8±11.6
		test	80.0	5.1	83.3	34.4
EPENet	0.25	cross-val	97.5±5.0	19.2±13.4	71.8±19.1	45.7±11.5
		test	80.0	12.8	77.8	45.7
EPENet	**0.30**	cross-val	**95.0** ± **10.0**	**26.8** ± **8.8**	**64.4 ± 21.6**	**54.6 ± 8.1**
		test	**80.0**	**28.2**	**61.1**	**58.3**
EPENet	**0.35**	cross-val	85.0±20.0	33.2±13.8	59.3±21.3	63.2±7.7
		test	**50.0**	**41.0**	**55.6**	**67.8**
EPENet	0.40	cross-val	81.0±18.5	39.9±15.5	43.2±31.7	71.2±8.8
		test	50.0	51.3	50.0	77.2
EPENet	0.45	cross-val	74.5±21.2	51.5±23.3	31.4±24.6	79.0±8.7
		test	40.0	61.5	38.9	83.7
EPENet	0.50	cross-val	68.7±19.8	54.3±22.4	25.7±25.0	82.6±6.8
		test	40.0	74.4	27.8	90.9
EPENet	0.55	cross-val	56.6±16.2	65.0±21.6	19.7±22.7	87.6±6.0
		test	40.0	87.2	27.8	95.7
EPENet	0.60	cross-val	53.8±11.3	79.6±14.0	17.9±17.2	92.1±3.9
		test	40.0	89.7	16.7	96.7
Radiologists	—	test	**50.0**	**76.9**	—	–

## Data Availability

The data presented in this study are available on request from the corresponding author.

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
