# Peer review of "Computational Detection of Extraprostatic Extension of Prostate Cancer on Multiparametric MRI Using Deep Learning"

_cancers, 2022, doi:10.3390/cancers14122821_

Round 1

Reviewer 1 Report

Although the aim of work was clearly to provide an automatic solution for the EPE detection the resulting specificity is so low that it could be interesting to test the use of the system as an aided detection tool and measure the outocome of the combination Radiologist+system and system + Radiologist. That is to let the radiologist revise his diagnosis once informed on the Sw results, or make the diagnosi after been told the sw results.

Please stress, in the discussion, the potential hazards of false positive outcome that derives as a result of the sw analysis

Reviewer 2 Report

The authors should be congratulated to their work on using deep learning models tot predict EPE on MR images of the prostate. There is a high need for better prediction of EPE and this study demonstrates the advances they have made. 

Below find some comments:

- At the end of the introduction there are parts that belong to the method and results section. These should be moved. Actually, the last paragraph could be completely removed

- 2.3 deep learning models might be moved to an appendix

- EPE prediction is dichotomized in yes or no. This makes the model more easy to use. In clinical practice the extend of EPE could be incorporated in the outcome (eg minor EPE (<1 mm) or >1 mm, etc). This might be of interest for future research

- in the final results the specificity on patient level is very low. This is well discussed but is rather disappointing. The strong part is the sensitivity which is higher than the radiologist and in this way it can be helpfull in EPE detection. But together with the low specificity this will result in a high number of false positives

Reviewer 3 Report

The manuscript entitled "Computational Detection of Extraprostatic Extension of Prostate Cancer on Multiparametric MRI Using Deep Learning" by S. Moroianu et al. presents an original study applying deep learning techniques to automatically assess EPE based on MRI images. Although the study does not report own innovative creations, authors clearly claimed that the goal is to evaluate existing deep learning methods as applied to the specific scopes. The manuscript is well organized, its redaction is clear and can be easily followed. Methodologies are appropriately described and conclusions are supported.

I have a minor suggestion regarding the significance of the alpha parameter. Authors comment that alpha = 0.30 appears as suitable, and data summarized in Table 4 support such a proposal. However, the selected variation step (0.1) may appear as somewhat large, thus providing a recommended range, instead of recommended value, for alpha would provide a better fit with the overall behavior, sensitivity, in terms of the alpha parameter.  

Reviewer 4 Report

Extraprostatic extension of disease (EPE) has a role in risk stratification of
prostate cancer patients, since it represent one of the major risk factor for biochemical recurrence after radical prostatectomy.

Recently, multiparametric MRI (mpMRI) has emerged as one of the detection of EPE. However, correctly identifying EPE through the acquired images is highly dependent on the radiologist's experience, with a high risk of false negative results.

Moroianu and colleagues, design a fully automated approache in order to improve the accuracy of EPE detection on mpMRI. They used deep learning models trained though post-surgical histological sections in combination of clinical createria of EPE indetification.

They observed that the deep learning approach shows a lower false negative detection rate than the radiologist's observations.

On the other hand, this method showed an higher rate of false positive results.

Although the first point is an important feature for correcting potential clinical misleading and reducing discrepancies between different radiologists and hospitals, the second point risks increasing the already existing problem of overdiagnosis and overtreatment.

Although the authors highlight several limitations of their study, they conclude that their approach is better than others as it is fully automated/automatable. They must clearly underline the advantages of their approach compared to the others already published, as the diagnostic performances appear much more limited than the data already present in the literature.

Round 2

Reviewer 4 Report

This reviewer appreciates the efforts of the authors